# VP-MonoMF: Visual Prompt Guided Monocular 3D Object Detection with Multiscale Fusion

## Abstract

Depth estimation from a single image remains a challenging task in monocular 3D object detection. Existing methods improve the detection accuracy by leveraging more precise 2D and 3D information. However, they simultaneously train 2D and 3D detection branches, which inevitably affect each other. Meanwhile, they often overlook the adverse effects caused by variations in camera pose. Furthermore, although they achieve satisfactory detection accuracy on large objects, their accuracy on small objects remains limited due to limited pixel areas. To address these issues, we propose a Visual Prompt Guided Monocular 3D Object Detection Method with Multiscale Fusion (VP-MonoMF). Specifically, we first develop a Multi-Depth Fusion (MDF) module as the 3D detection branch, which integrates multi-scale information from both global depth maps and local 3D depth information. Then, we train MDF in the first stage and the 2D Detector in the second stage to mitigate mutual interference. To minimize the impact of the camera pose variance, MDF utilizes a 3D Depth Reconstruction (3DR) module to correct depth map deviations. Furthermore, we introduce a Visual Prompt Fusion (VPF) module to enhance small object features by adaptively adjusting weights based on object size. We conduct experiments on the KITTI dataset. VP-MonoMF achieves state-of-the-art performance in monocular 3D object detection task. The code will be made available upon acceptance of the paper.

## 1 INTRODUCTION

3D object detection identifies and locates objects in a three-dimensional space using computer vision techniques. It can pinpoint the spatial coordinates and orientations of objects using their depth information in the real world. With the development of advanced technologies such as machine learning and LiDAR, 3D object detection has become fundamental for machines to understand the physical environment. For example, it has been widely used in autonomous driving (Mao et al., 2022; Ma et al., 2022) and robot navigation (Chaturvedi et al., 2024; Wijesekara, 2022).

Monocular 3D object detection has attracted widespread attention due to its lower cost and simple configuration compared to other 3D object detection methods (Zhang et al., 2024). It estimates the 3D information from a single 2D image based on 2D and 3D detection branches. Recent monocular 3D object detection methods can be divided into two groups: image based and image with extra information based. Image based methods (Yan et al., 2024a; Zhu et al., 2023) only utilize a single RGB image captured by the monocular camera for depth estimation. For image with extra information based methods (Huang et al., 2024), they further utilize prior knowledge or auxiliary information to improve the detection accuracy. Although these methods reduce the complexity, it is still challenging to guarantee the performance.

First, the 2D and 3D detection branches share the same backbone for image feature extraction and they are trained simultaneously. Unfortunately, the 2D and 3D detection branches inevitably affect each other, which introduces non-negligible noise for 3D detection branches (Liu et al., 2020). Second, current monocular 3D object detection methods only consider the scenarios with a fixed camera. However, the camera position may change due to the surrounding environment. As shown in Figure 1, the camera is unstable because of vibration. Thus, it leads to deviations in depth map

Figure 1: Impact of the environment on the camera pose. As shown in the left image, the camera maintains a fixed position on a robot car when the surrounding environment does not change. The right figure shows that the camera position changes due to the vibrations caused by the environment.

estimation. Third, images captured by monocular cameras usually include small objects that occupy limited areas. Thus, it is difficult to estimate the depth due to insufficient features.

Facing these challenges, we propose a Visual Prompt Guided Monocular 3D Object Detection Method with Multiscale Fusion (VP-MonoMF). It consists of three modules: Multi-Depth Fusion (MDF), 3D Depth Reconstruction (3DR), and Visual Prompt Fusion (VPF). In the first stage, the MDF module first estimates the depth map and the dimensions of objects. Then the dimensions and the 2D height from ground truth (GT) are used to estimate the 3D depth information. Meanwhile, the 3DR module reconstructs the depth map based on the camera pose estimation. Finally, we fuse the reconstructed depth map with the 3D depth information to obtain the accurate depth. In the second stage, we freeze the MDF module and train the 2D Detector using the enhanced features from VPF. Then we combine the outputs of the MDF and 2D Detector to get 3D detection results.

Specifically, for the first challenge, we build an MDF module based on the depth and dimension detectors. The detectors generate a global depth map and local 3D depth information of objects, which are further refined and fused to build an accurate depth. Then we train the MDF module and the 2D detector in different stages to avoid the influence. For the second challenge, we design a 3DR module based on the camera pose variance for the 3D detection branch. First, the camera transformation matrix that reflects the camera pose variance is obtained by the estimated vanishing point and horizon information of images. Then, we use it to correct the depth map. For the third challenge, we train the 2D Detector in the second stage and design a VPF module for the 2D detector to optimize the detection performance of the object's 2D properties. We first convolve the features to get the attention map. Meanwhile, we generate a visual prompt to adaptively adjust the attention map according to the size of objects from GT. Finally, we use the adjusted attention map to enhance the object features for different image areas. Thus, the 2D detector can better extract the features of objects.

The contributions of this paper are summarized as follows:

- We propose a 3D detection module MDF to fuse the global depth map and the local 3D depth information of each object. This module is trained in separate stages from the 2D detector.

- We propose a 3DR module considering the camera position and orientation. It utilizes the camera transformation matrix to correct the depth map and effectively reduces the deviation in the depth map.

- We propose a VPF module based on the visual prompt. The visual prompt adjusts the attention map dynamically. To the best of our knowledge, this is the first work that explores GT-based visual prompt for the task of monocular 3D object detection.

- Experiments demonstrate that our method achieves state-of-the-art performance on the KITTI 3D detection benchmark without using additional data.

## 2 RELATED WORK

### 2.1 IMAGE BASED METHODS

Image based methods (Lin et al., 2024; Zhang et al., 2025) estimate 3D and 2D information of objects from RGB images instead of external data or pre-trained models. The estimated 2D and 3D

information are combined to obtain the 3D bounding boxes. For example, MonoPGC (Wu et al., 2023) introduced pixel depth estimation as the auxiliary task and designed a depth cross-attention pyramid module to inject local and global depth geometry knowledge into visual features. By incorporating depth information into the masking process, MonoMAE (Jiang et al., 2024a) enhanced feature representation, enabling the model to better capture spatial relationships and object geometry. WeakMono3D (Tao et al., 2023) incorporated projection and multi-view consistencies to guide the prediction of 3D bounding boxes by two consistency losses. They also proposed a 2D direction label to replace the 3D rotation label marked on the point cloud data. Although these methods improve the accuracy and robustness of monocular 3D object detection, they still have limitations in complex scenarios and detecting incomplete objects.

## 2.2 IMAGE WITH EXTRA INFORMATION BASED METHODS

Image with extra information based methods (Li et al., 2024b; Choi et al., 2024) utilize extra information to help the model better understand the 3D information of objects, which includes pre-trained models, annotated keypoints, and Computer-Aided Design (CAD) models.

The pre-trained models estimate the extra information for 3D object detection. For example, MonoNeRD (Xu et al., 2023) used a Neural Radiance Fields model to enable accurate 3D perception and employ volume rendering to recover RGB images and depth maps. YOLOBU (Xiong et al., 2024) used a Deformable DETR model with cross-attention mechanism to build the connections of pixels for detection. Although pre-trained models can obtain more accurate information to help improve the detection performance, they are highly complex.

The annotated keypoints guide and supervise the model estimation results. For example, LPCG (Peng et al., 2022) generated pseudo labels from unlabeled LiDAR point clouds which can be applied for any monocular 3D detector to use massive unlabeled data. OVM3D (Huang et al., 2024) automatically combined images with 3D object labels to utilize internet-scale data. However, they require more annotations for training and are usually less generalized.

CAD models provide accurate 3D shape information for the network. MonoGRK (Barabanau et al., 2019) combined region-based detectors and a geometric reasoning step over keypoints using real-world images and CAD models. AutoShape (Liu et al., 2021) automatically fitted the 3D shape to the visual observations and then generated GT annotations of 2D/3D keypoint pairs for the network. Although CAD models help improve inspection performance, they have slow inference speeds.

## 3 METHOD

### 3.1 OVERVIEW

Figure 2 shows the architecture of VP-MonoMF, a visual prompt Guided monocular 3D object detection method with multiscale fusion. In the first stage, we focus on extracting depth information from the monocular image $I_{in}$ and training the MDF module. $I_{in}$ is input into the backbone network Deep Layer Aggregation (DLA) (Yu et al., 2017) to obtain the feature $F \in R^{W \times H \times C}$, where $W$ and $H$ are the width and height of the feature, and $C$ is the number of channels. The 2D height from GT is denoted as $height^*$. Then $height^*$ and $F$ are input into MDF to obtain the fused depth $Z_{com}$ and the dimension $dim$ of objects.

In the second stage, we also use DLA to get the feature $F$. Then we use the VPF module to enhance object features. It helps the 2D detector to estimate the 2D properties. Specifically, VPF dynamically generates a visual prompt to enhance the feature $F$, and the enhanced feature $F_{vp} \in R^{W \times H \times C}$ is input to the 2D Detector. The 2D Detector generates 2D properties *offset*, *center*, *orientation* and *height* and we input *height* to the MDF module for $Z_{com}$. Then, we combine the outputs of the 2D Detector and the MDF module to generate the object's 3D bounding boX. We freeze the DLA and MDF modules, and train only VPF and the 2D Detector in this stage. Note that only the second stage is used for testing.

### 3.2 MDF

We design a multiscale depth fusion module MDF. This module fuses the global depth map $Z_{glo}$ and local depth $Z_{loc}$ for a comprehensive depth $Z_{com}$ utilizing a two-branch architecture. It increases the accuracy of the estimated depth to detect objects.

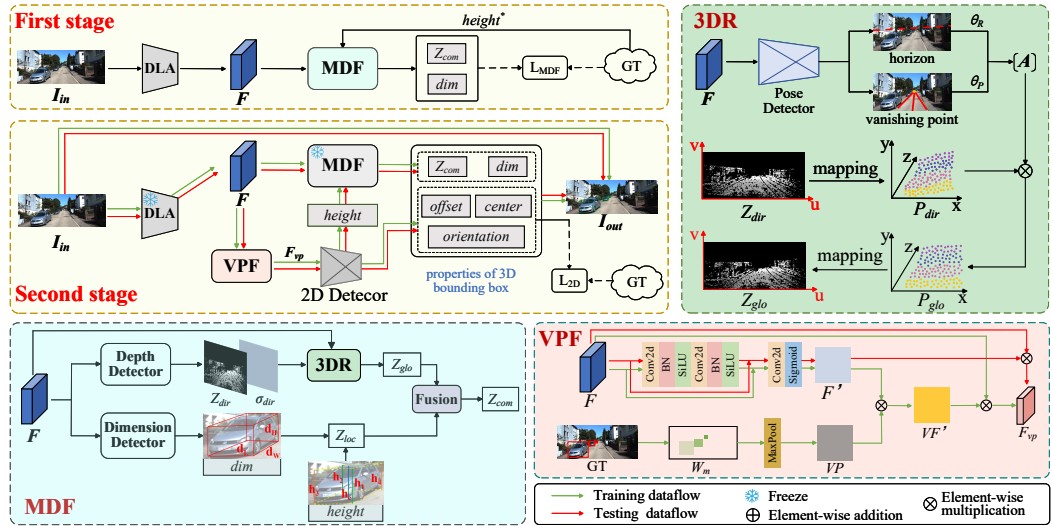

Figure 2: The framework of VP-MonoMF. Our method consists of two stages. The training and testing processes of the second stage are represented by green and red arrows, respectively. In the first stage, we train the MDF module and backbone DLA to get accurate depth. The 3DR module in MDF reconstructs the depth map. In the second stage, we train the 2D Detector and VPF to avoid mutual influence. The enhanced features generated by VPF are fed into the 2D Detector. The detection result $I_{out}$ is obtained by combining the outputs of the 2D Detector and MDF module.

In the first branch, we get $Z_{glo}$ from the 3DR module. Specifically, we input $F$ into the Depth Detector to obtain an estimated depth map $Z_{dir} \in R^{W' \times H'}$ and the corresponding reliability score $\sigma_{dir} \in R^{W' \times H'}$, where $W'$ and $H'$ are the width and height of the $Z_{dir}$ and $\sigma_{dir}$. $\sigma_{dir}$ reflects the confidence of the estimated depth. Then we input $Z_{dir}$ and $F$ into the 3DR module to get $Z_{glo}$. Note that we also have the reliability score $\sigma_{glo}$ for $Z_{glo}$ and it is set equal to $\sigma_{dir}$. We introduce the 3DR module in the following section.

In the second branch, we get $Z_{loc}$ from the 3D property $dim$ and the 2D $height$. $dim$ includes the object's 3D properties: length, width, and height. Specifically, we first input $F$ into the Dimension Detector to obtain $dim = \{(d_L, d_W, d_H)_i | i = 1, 2, .., N\}$, where $N$ is the number of objects. Meanwhile, we get 2D $height$ from the 2D detector or GT. It contains the projected height $h_i (i = 1, 2, 3, 4, c)$ of four vertical edges of the bounding box, and $h_c$ is for the center line. Their reliability score is $\sigma_i (i = 1, 2, 3, 4, c)$. We use $d_H$ of the object and its corresponding projected height $h_i$ to calculate the depth of four vertical edges and the center line (Cai et al., 2020):

$$z_i = \frac{f \times d_H}{h_i} , \tag{1}$$

where $f$ is the focal length of the camera. To increase the estimation accuracy of the center line, we calculate two average depths $z_{d1}$ and $z_{d2}$ based on the depths of four vertical edges $z_i (i = 1, 2, 3, 4)$:

$$z_{d1} = \frac{z_1 + z_2}{2}, z_{d2} = \frac{z_3 + z_4}{2} . \tag{2}$$

Similarly, we calculate the reliability scores $\sigma_{d1}$ and $\sigma_{d2}$:

$$\sigma_{d1} = \frac{\sigma_1 + \sigma_2}{2}, \sigma_{d2} = \frac{\sigma_3 + \sigma_4}{2} . \tag{3}$$

Thus, we have $Z_{loc} = \{z_{d1}, z_{d2}, z_c\}$ for the depth of the object center $(x', y')$.

We also get the depth $z_{x'y'}$ from $Z_{glo}$ of the center $(x', y')$ and the corresponding reliability score $\sigma_{x'y'}$. Then, we allocate different weights for $z_{d1}, z_{d2}, z_c$, and $z_{x'y'}$ based on their reliability scores to get a more accurate depth $Z_{com}$. The formula is as follows:

$$Z_{com} = \frac{\sum\limits_{k \in \{d1, d2, c, x'y'\}} z_k \times \sigma_k}{\sum\limits_{k \in \{d1, d2, c, x'y'\}} \sigma_k} , \tag{4}$$

where $\sigma_k \in (0, 1]$.

## 3.3 3DR

Camera pose variance may lead to the camera's optical axis not being parallel to the ground, which introduces deviations when estimating the depth map. The 3DR module corrects the deviations using a camera transformation matrix and reconstructs the depth map based on the camera projection.

We first convert $Z_{dir}$ into points $P_{dir}$ in the camera coordinate system:

$$x = \frac{(u - c_u)z_{uv}}{f}, y = \frac{(v - c_v)z_{uv}}{f} , \tag{5}$$

where $(u, v)$ is the pixel coordinate in $Z_{dir}$; $z_{uv}$ is the depth value at $(u, v)$; $x$ and $y$ are the coordinates of the camera coordinate system; $(c_u, c_v)$ is the center coordinate of the image. The center coordinate and focal length are intrinsic camera parameters.

Meanwhile, we input $F$ into the Pose Detector Zhou et al. (2022) to obtain the horizon and vanishing point. The horizon is a straight line where the ground and sky meet in the image. It is represented by the linear equation $y = ax + b$. The vanishing point is the point where the parallel road boundaries converge. Its coordinate is $(x_{vp}, y_{vp})$. Then, we get the roll angle $\theta_R$ and pitch angle $\theta_P$ of the camera from the horizon and vanishing point:

$$\theta_R = \arctan(a), \theta_p = \arctan(\frac{x_{vp} - c_u}{f}) . \tag{6}$$

Then we get the transformation matrix $A$ based on $\theta_R$ and $\theta_P$:

$$A_R = \begin{bmatrix} cos\theta_R & -\sin\theta_R & 0 \\ \sin\theta_R & \cos\theta_R & 0 \\ 0 & 0 & 1 \end{bmatrix}, A_P = \begin{bmatrix} 1 & 0 & 0 \\ 0 & \cos\theta_P & -\sin\theta_P \\ 0 & \sin\theta_P & \cos\theta_P \end{bmatrix} , \tag{7}$$

$$A = A_R A_P . \tag{8}$$

After we get $A$ and $P_{dir}$, we use $A$ to convert $P_{dir}$ to $P_{glo}$. The formula is as follows:

$$(\bar{x}, \bar{y}, \bar{z}_{u'v'}) = (x, y, z_{uv})A , \tag{9}$$

where $(x, y, z_{uv})$ is the coordinate of $P_{dir}$ and $(\bar{x}, \bar{y}, \bar{z}_{u'v'})$ is the coordinate of $P_{glo}$ after correction.

Finally, we convert $P_{glo}$ to $Z_{glo}$ using formula (5).

## 3.4 VPF

The VPF module enhances object features using the visual prompt. As shown in Figure 2, the visual prompt is a mask that reflects the distribution of feature attention. Note that the visual prompt is only used during training to facilitate the training of the 2D Detector.

We obtain the attention map $F' \in R^{W \times H}$ through dual Conv-BN-SiLU paths with a skip connection and a Conv-Sigmoid layer. Meanwhile, we obtain object sizes from the 2D bounding boxes in GT. Then we employ a learnable sigmoid-based weighting mechanism to assign weights $\omega_i \in (0, 1)$ based on object sizes:

$$w_i = \frac{1}{1 + e^{(\beta \cdot s_i - b)/T}} , \tag{10}$$

where $S_i \in (0, +\infty)$ represents the size of the $i$-th object. $\beta \in (0, +\infty)$ indicates the sensitivity to object size, where a larger value emphasizes smaller objects. $b \in (0, +\infty)$ adjusts the weight baseline and a positive value elevate the weight distribution. $T \in (0, +\infty)$ is initialized to 1.0. A larger value generates more uniform weight distributions. Note that $\beta$, $b$, and $T$ are learnable parameters.

We create a visual prompt mask $W_m \in R^{\hat{W} \times \hat{H}}$, where $\hat{W}$ and $\hat{H}$ are consistent with the width and height of the input image $I_{in}$. We assign $1 + w_i$ to each area covered by the 2D bounding box of the $i$-th object, and set the area not covered by objects to 1 to preserve its attention values. For areas with multiple objects, we use the maximum value of $w_i$. We obtain a visual prompt $VP \in R^{W \times H}$ by performing a maximum pooling on $W_m$ to be consistent with the size of $F'$.

Then we multiply $F'$ and $VP$ to obtain the adjusted attention map $VF' \in R^{W \times H}$. Finally, we multiply $VF'$ with the feature $F$ to obtain the enhanced feature $F_{vp}$. Note that during testing, we do not need a visual prompt and there is no GT in the testing dataset. Thus, $F_{vp}$ is obtained by multiplying $F'$ and $F$, where $F'$ serves as the attention map.

### 3.5 Loss Function

The loss function for the first stage is denoted as $L_{MDF}$. It is defined as follows:

$$L_{MDF} = L_{dir} + L_{\dim} + L_{pose} \, . \tag{11}$$

$L_{dir}$ represents the $L_1$ loss of the Depth Detector in the MDF module. It is defined as follows:

$$L_{dir} = |Z_{dir} - Z^*| \times \sigma_{dir} + \log(\frac{1}{\sigma_{dir}}) \, , \tag{12}$$

where $Z^*$ represents the depth map of GT.

$L_{dim}$ represents the $L_1$ loss of the Dimension Detector in the MDF module. It is defined as follows:

$$L_{\dim} = \sum_{k \in \{H, W, L\}} |d_k - d_k^*| \, , \tag{13}$$

where $d_k^*$ represents the 3D height, width, and length properties of objects from GT.

$L_{pose}$ represents the loss of the Pose Detector in the 3DR module. It is defined as follows:

$$L_{pose} = ||A - A^*||_F \, , \tag{14}$$

where $A^*$ represents the transformation matrix of GT and $|| \cdot ||_F$ represents the Frobenius norm.

The loss function for the second stage is $L_{2D}$, which constrains the 2D Detector to learn the *offset*, *center*, *orientation*, and *height* properties. It is defined as follows:

$$
\begin{aligned}
L_{2D} = L_{cen} + L_{off} + L_{hei} + L_{ori} \quad &= \frac{1}{N} \sum_{i=1}^{N} (|x_i - x_i^*| + |y_i - y_i^*|) \\
&+ \frac{1}{N} \sum_{i=1}^{N} (|o_i^x - o_i^{x*}| + |o_i^y - o_i^{y*}|) \\
&+ \frac{1}{N} \sum_{i=1}^{N} |h_i - h_i^*| \cdot \sigma_i + \log(\sigma_i) + \frac{1}{N} \sum_{i=1}^{N} |\theta_i - \theta_i^*| \, ,
\end{aligned}
\tag{15}
$$

where $L_{cen}$ represents the center loss, $L_{off}$ represents the offset loss, $L_{hei}$ represents the height loss, and $L_{ori}$ represents the orientation loss; $h_i^*$ is the GT of the 2D *height*, $\sigma_i$ is the reliability score generated by the 2D Detector; $(x_i^*, y_i^*)$, $(o_i^{x*}, o_i^{y*})$, and $\theta_i^*$ represent the center coordinates, offset and the orientation angle from GT, respectively.

## 4 Experiment

### 4.1 Experimental Setup

**Datasets and Metrics.** Our experiments are conducted on the widely used KITTI dataset (Geiger et al., 2012). The dataset includes 7,481 annotated images, split into a training set (3,712 images) and a validation set (3,769 images). It also has a separate test set of 7,518 images. Objects are categorized into three difficulty levels: Easy, Moderate (Mod), and Hard, which are determined by factors such as the height of bounding boxes, occlusion, and truncation. KITTI also provides evaluation protocols including Average Precision (AP) for 3D detection and bird's eye view (BEV) detection. We evaluate the performance using $AP_{3D}$ and $AP_{BEV}$ for the 3D bounding box and BEV, respectively. We focus on the car category with *easy*, *mod*, and *hard*. To facilitate comparison with previous studies, we report detection results with an IoU threshold of 0.7 for the car category. Note that we also have results on the nuScenes dataset in the supplementary material.

Table 1: $AP_{3D}$ and $AP_{BEV}$ of different methods on the KITTI test set.

| Methods | Extra data | Test — $AP_{3D}$(%) | | | Test — $AP_{BEV}$(%) | | |
|---|---|---|---|---|---|---|---|
| | | Easy | Mod. | Hard | Easy | Mod. | Hard |
| AutoShape (Liu et al., 2021) *ICCV'21* | CAD | 22.47 | 14.17 | 11.36 | 30.66 | 20.08 | 13.10 |
| DCD (Li et al., 2022) *ECCV'22* | | 23.81 | 15.90 | 13.21 | 32.55 | 21.50 | 18.25 |
| MonoRun (Chen et al., 2021) *CVPR'21* | LiDAR | 19.65 | 12.30 | 10.58 | 27.94 | 17.34 | 15.24 |
| MonoDTR (Huang et al., 2022a) *CVPR'22* | | 21.99 | 15.39 | 12.73 | 28.59 | 20.38 | 17.14 |
| SMOKE (Liu et al., 2020) *CVPR'20* | None | 14.03 | 9.76 | 7.84 | 20.83 | 14.49 | 12.75 |
| MonoPair (Chen et al., 2020) *CVPR'20* | | 16.28 | 12.30 | 10.42 | 24.12 | 18.17 | 15.76 |
| MonoDLE (Ma et al., 2021) *CVPR'21* | | 17.23 | 12.26 | 10.29 | 24.79 | 18.89 | 16.00 |
| MonoFlex (Zhang et al., 2021) *CVPR'21* | | 19.94 | 13.89 | 12.07 | 28.23 | 19.75 | 16.89 |
| MonoCon (Liu et al., 2022a) *AAAI'22* | | 22.50 | 16.46 | 13.95 | 31.12 | 22.10 | 19.00 |
| MonoGround (Qin & Li, 2022) *CVPR'22* | | 21.37 | 14.36 | 12.62 | 30.07 | 20.47 | 17.74 |
| MPMonoD (Shi et al., 2023) *WACV'23* | | 20.08 | 13.72 | 11.34 | - | - | - |
| GRAMO (Guan et al., 2024) *FC'24* | | 22.34 | 15.67 | 13.12 | 32.44 | 21.74 | 18.38 |
| MonoCD (Yan et al., 2024a) *CVPR'24* | | 25.53 | 16.59 | 14.53 | 33.41 | 22.81 | 19.57 |
| MonoMAE (Jiang et al., 2024a) *NeurIPS'24* | | 25.60 | 18.84 | 16.78 | 34.15 | 24.93 | 21.76 |
| MonoDGP (Zhang et al., 2025) *CVPR'25* | | **26.35** | 18.72 | 15.97 | **35.24** | 25.23 | 22.02 |
| **VP-MonoMF(Ours)** | None | 25.81 | **18.92** | **16.92** | 35.15 | **25.36** | **22.67** |

**Implementation Details.** We implement our method based on DLA34 (Yu et al., 2017) following the settings in Yan et al. (2024a). The input image resolution is $1280 \times 384$. The feature map of the backbone is $320 \times 96 \times 64$. The Depth Detector, Dimension Detector, and 2D Detector attached to the backbone consist of one Conv $(3 \times 3 \times 256)$-BN-ReLU and another Conv $(1 \times 1 \times C')$ layer, where $C'$ is the output channel. In the training stage, we use the Adam optimizer with a batch size of 8 for 100 epochs. The initial learning rate is $3 \times 10^{-4}$ and the decay weight is $1 \times 10^{-5}$. We run the experiments on a PC with a single RTX 4090 GPU.

## 4.2 COMPARISON WITH STATE-OF-THE-ART METHODS

Table 1 shows the $AP_{3D}$ and $AP_{BEV}$ obtained on the KITTI test dataset. Note that the best results are in **bold** and the second-best are underlined. Compared to all the methods, our method achieves the best performance except for the *easy* objects in $AP_{3D}$. Specifically, compared with the Mono-MAE method, our method increases the $AP_{3D}$ by 0.21%, 0.08%, and 0.14% for *easy*, *mod*, and *hard*, respectively. The $AP_{BEV}$ also increases by 1%, 0.43%, and 0.91%. In addition, our method achieves 0.2%/0.13%, and 0.95%/0.65% improvement in $AP_{3D}/AP_{BEV}$ than the state-of-the-art MonoDGP method for *mod* and *hard*. The results demonstrate that VP-MonoMF benefits from fusing multiple depths and the visual prompt for 3D detection.

## 4.3 ABLATION STUDY

We verify the effectiveness of each module, the contribution of different depths, and the utility of the two-stage strategy and visual prompt on the KITTI.

**Contribution of each module.** Table 2 shows the contribution of each module, where "Baseline" means that we do not use MDF, 3DR, and VPF modules. "MDF-" means that the 3DR module is not included. The third line indicates that the MDF module achieves 4.92%/5.27%, 4.83%/4.56%, and 3.99%/5.11% improvement in $AP_{3D}/AP_{BEV}$ on three levels of difficulty compared with "Baseline". This indicates that our 3D detection branches generate an accurate depth. In addition, the 3DR module improves a significant 1.59%/3.28%, 2.73%/2.32%, and 2.37%/2.45% in $AP_{3D}/AP_{BEV}$ compared with "MDF-". It highlights the importance of reconstructing the depth map. When adding the VPF module, it contributes to 2.9%/1.86%, 2.36%/2.29%, and 2.56%/1.58% improvement in $AP_{3D}/AP_{BEV}$, which demonstrates the effectiveness of the VPF module.

**Contribution of different depths.** Table 3 shows the contribution of different depths. $Z_{loc}$, $Z_{dir}$, $Z_{glo}$, and $Z_{com}$ indicate that we use them separately as outputs of the MDF module. We find that $Z_{loc}$ and $Z_{dir}$ have minor performance differences. However, compared to $Z_{dir}$, $Z_{glo}$ significantly achieves 3.4%/3.85%, 2.62%/2.48%, and 1.85%/1.14% improvement in $AP_{3D}/AP_{BEV}$. It

Table 2: Contribution of each module.

| Baseline | MDF- | 3DR | VPF | Val — $AP_{3D}(\%)$ | | | Val — $AP_{BEV}(\%)$ | | |
|---|---|---|---|---|---|---|---|---|---|
| | | | | Easy | Mod. | Hard | Easy | Mod. | Hard |
| ✓ | × | × | × | 22.60 | 13.74 | 11.19 | 31.45 | 21.95 | 18.32 |
| ✓ | ✓ | × | × | 25.93 | 15.84 | 12.81 | 33.44 | 24.19 | 20.98 |
| ✓ | ✓ | ✓ | × | 27.52 | 18.57 | 15.18 | 36.72 | 26.51 | 23.43 |
| ✓ | ✓ | ✓ | ✓ | **30.42** | **20.93** | **17.74** | **38.58** | **28.80** | **25.01** |

Table 3: Contribution of different depths.

| Depth | Val — $AP_{3D}(\%)$ | | | Val — $AP_{BEV}(\%)$ | | |
|---|---|---|---|---|---|---|
| | Easy | Mod. | Hard | Easy | Mod. | Hard |
| $Z_{loc}$ | 22.92 | 15.46 | 13.75 | 30.56 | 23.49 | 21.11 |
| $Z_{dir}$ | 23.51 | 16.24 | 14.32 | 31.96 | 23.89 | 22.62 |
| $Z_{glo}$ | 26.91 | 18.86 | 16.17 | 35.81 | 26.37 | 23.76 |
| $Z_{com}$ | **30.42** | **20.93** | **17.74** | **38.58** | **28.80** | **25.01** |

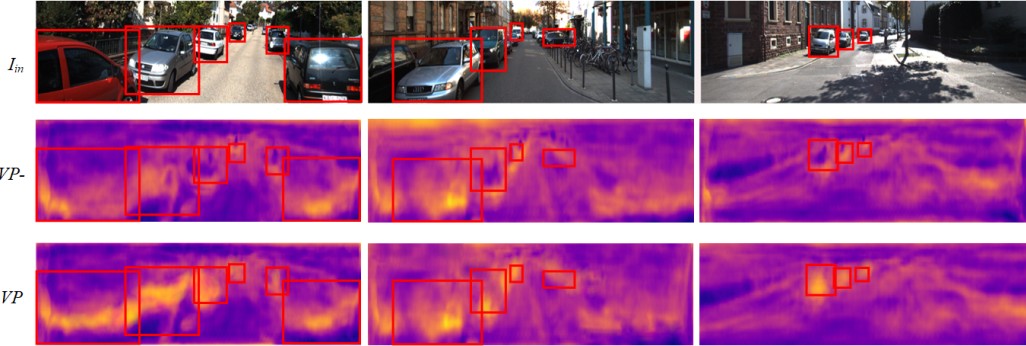

Figure 3: The heatmaps without and with the visual prompt. The visual prompt focuses on the area where objects are located. The red rectangle represents the 2D bounding box of the object. The brighter color indicates a higher attention score. $VP$-/$VP$ represents the original attention map without/with a visual prompt, respectively.

shows the effectiveness of 3DR. In addition, $Z_{com}$ contributes to 3.51%/2.77%, 2.07%/2.43%, and 1.57%/1.25% improvement in $AP_{3D}/AP_{BEV}$ compared with $Z_{glo}$, which proves the effectiveness of depth fusion.

**Contribution of two-stage and visual prompt.** Table 4 shows the impact of the two-stage and the visual prompt. Without the visual prompt indicates that we only include the testing data flow in the VPF module. One-stage refers to training 3D detection branches and 2D detection branches simultaneously. Specifically, we train them in the second stage without freezing the DLA and MDF modules. We find that the performance of the one-stage is lower than that of the two-stage on all levels of difficulty. It indicates that the two-stage pipeline reduces the mutual impact during training. We also find that the visual prompt improves a significant 2.12%/1.43%, 1.78%/1.68%, and 1.98%/1.15% in $AP_{3D}/AP_{BEV}$, which justifies the effectiveness of the visual prompt.

### 4.4 QUALITATIVE RESULTS

Figure 3 shows the visualization of the visual prompt. $I_{in}$ is the input image. $VP$-/$VP$ is the original attention map without/with a visual prompt, respectively. We use heatmaps to reflect the attention in the image, in which a brighter color indicates higher attention scores. We find that $VP$ has higher attention for objects, especially for small objects. As shown in Figure 3, the color of small targets is brighter than $VP$-, which validates the effectiveness of the visual prompt.

Figure 4 shows the 3D detection bounding boxes and the BEV obtained from different estimated depths. $S_{loc}$, $S_{dir}$, $S_{com}$ mean that we use $Z_{loc}$, $Z_{dir}$, $Z_{com}$ as the depth, respectively. $S_{loc}+S_{dir}$ means using fused $Z_{loc}$ and $Z_{dir}$ as the depth. We find a large deviation between the predicted box

Table 4: Contribution of the two-stage and the visual prompt.

| Stage | Visual prompt | Val — $AP_{3D}(\%)$ | | | Val — $AP_{BEV}(\%)$ | | |
|---|---|---|---|---|---|---|---|
| | | Easy | Mod. | Hard | Easy | Mod. | Hard |
| One-stage | w/o | 25.64 | 17.12 | 13.65 | 34.08 | 24.71 | 20.76 |
| Two-stage | w/o | 28.30 | 19.15 | 15.76 | 37.15 | 27.12 | 23.86 |
| | w | **30.42** | **20.93** | **17.74** | **38.58** | **28.80** | **25.01** |

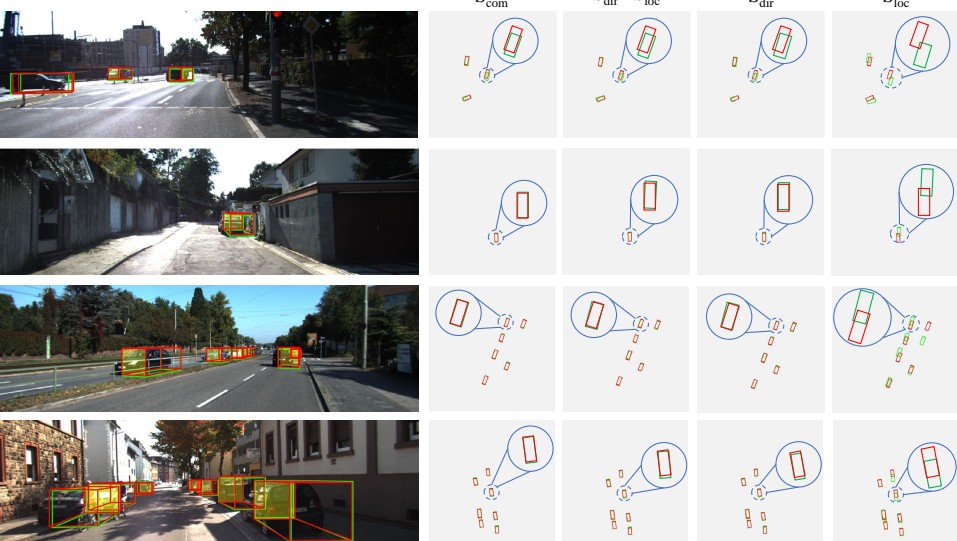

Figure 4: Qualitative examples on the KITTI validation set. Each row displays a 3D scene detection result along with four BEV visualizations. The BEV visualizations reflect the differences in depth. From left to right: $S_{com}$, $S_{loc} + S_{dir}$, $S_{loc}$, $S_{dir}$. Red represents the GT of the box and green represents the prediction box.

and the GT when using $S_{loc}$ or $S_{dir}$. For example, the prediction box is obviously non-overlapped with the GT for $S_{loc}$ and $S_{dir}$. However, we find that $S_{loc} + S_{dir}$ reduces the deviation. When using $S_{com}$, the predicted box is closer to the GT. This visualizes the effectiveness of our depth fusion and the 3DR module.

## 5 CONCLUSION

In this paper, we propose a two-stage multiscale monocular 3D object detection method with a visual prompt. In the first stage, we train an MDF module that extracts depth information on a multiscale scale to enhance accuracy. In the second stage, we focus on training the 2D Detector with the enhanced features from VPF. In addition, we reconstruct a more accurate depth map by correcting the camera pose using the camera transformation matrix. To improve the performance on small objects, we use a visual prompt to enhance the features of the object area, which dynamically adjusts the feature enhancement. Extensive results demonstrate that our method achieves state-of-the-art results on the KITTI dataset.

**Limitations.** However, the performance on *easy* is not the best. This is because our method pays more attention to small objects. We believe that it can be improved by adjusting the weights of different objects. Meanwhile, the reliability scores estimated by the detectors may be biased, which affects the accuracy of the fused depth map. However, our method can also achieve satisfactory performance when using accurate reliability scores. In the future, we will focus on the aforementioned problems and apply it to scenarios where there are severe changes in the environment.

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

## SUPPLEMENTARY MATERIAL

## A  NETWORK ARCHITECTURE OF THE POSE DETECTOR

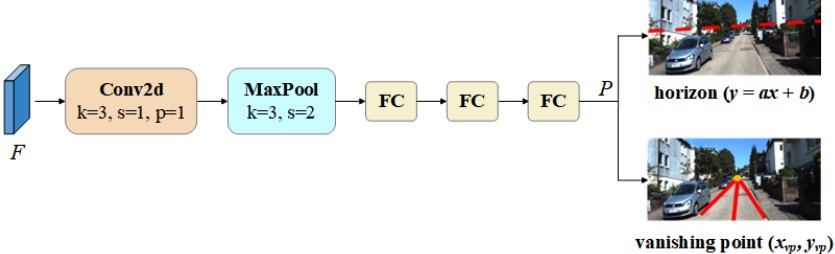

Figure 5: Structure of the Pose Detector.

In section of 3DR, we utilize the Pose Detector to obtain the horizon and vanishing point in 3DR. The detailed structure of the Pose Detector is shown in Figure 5. It consists of a $3 \times 3$ Convolution layer, a Max Pooling layer, and three Fully Connected layers. The input of the Pose Detector is the feature map $F$ from the backbone network DLA. The output vector $P = (a, b, x_{vp}, y_{vp})$, which is

Table 5: Inference Time, computation cost, and parameter size of VP-MonoMF

| Methods | Inference Time | FLOPs | Param. |
|---|---|---|---|
| MonoDETR (senrui Zhang et al., 2022) | 43ms | **62.12G** | - |
| GUPNet (Silberstein et al., 2016) | 40ms | 62.32G | - |
| MonoDTR (Huang et al., 2022b) | 37ms | 120.48G | - |
| MonoMAE (Jiang et al., 2024b) | 36ms | - | - |
| MonoCD (Yan et al., 2024b) | **19ms** | 142.89G | **16.52M** |
| MonoDGP (Pu et al., 2025) | 42ms | 68.99G | 38.90M |
| Ours | 20ms | 153.05G | 17.67M |

Table 6: Evaluation on the nuScenes validation set.

| Methods | mAP↑ | NDS↑ | mATE↓ | mASE↓ | mAOE↓ |
|---|---|---|---|---|---|
| CenterNet3D (Tang et al., 2020) | 0.306 | 0.328 | 0.716 | 0.264 | 0.609 |
| FCOS3D (Wang et al., 2021) | 0.343 | 0.415 | 0.725 | 0.263 | 0.422 |
| PETR (Liu et al., 2022b) | 0.370 | 0.442 | 0.711 | 0.251 | 0.433 |
| WeakPETRv2 (Han et al., 2024) | 0.375 | 0.421 | 0.809 | 0.272 | - |
| BEVFormer (Li et al., 2024a) | **0.416** | **0.517** | 0.673 | 0.274 | 0.372 |
| FCOS3D+MonoPlace3D (Parihar et al., 2025) | 0.370 | 0.440 | - | - | - |
| **Ours** | 0.394 | 0.464 | **0.645** | **0.247** | **0.364** |

$R^{1 \times 4}$. This Pose Detector has achieved state-of-the-art performance and has been widely used in 3D object detection (Chang et al., 2018; Zhou et al., 2025).

# B  EXPERIMENT

## B.1  EVALUATION OF RUNNING SPEED, COMPUTATION COST, AND PARAMETER SIZE

Table 5 shows the inference time, computation cost, and number of model parameters. We achieve better performance compared with MonoDGP in terms of inference time and parameters. Meanwhile, our performance is similar to MonoCD. Note that we achieve the best performance considering AP3D and APBEV for *mod* and *hard* in Table 1 of the paper. For *easy*, we achieve the second best performance.

## B.2  EVALUATION ON NUSCENES DATASET

To further prove the effectiveness of our method, we evaluate it on another popular dataset nuScenes. nuScenes comprises 1,000 video scenes, including RGB images captured by 6 surround-view cameras. The dataset has a training set (700 scenes), a validation set (150 scenes), and a test set (150 scenes). We report detection results on the validation set following the same setup (Tang et al., 2020; Wang et al., 2021; Liu et al., 2022b; Han et al., 2024) to facilitate comparison with previous studies. The performance of different methods is reported in Table 6.

Table 6 shows the mean Average Precision (mAP), nuScenes Detection Score (NDS), mean Average Translation Error (mATE), mean Average Scale Error (mASE) and mean Average Orientation Error (mAOE). mASE evaluates how accurately the dimension detector predicts the size of objects compared to their ground-truth annotations. Our method achieves the best performance in mASE due to the two-stage training framework which reduces the negative impact of the 2D detection branch on the dimension detector. mATE quantifies how well our method predicts the center position of detected objects compared to their ground-truth locations. mAOE quantifies the angular error between the predicted orientation and the true orientation of detected objects. We also achieve the best results on mAOE and mATE because our visual prompt effectively contributes to accurate center localization and orientation.

## B.3  MORE VISUAL RESULTS

This is an extension of the results in Section of Qualitative Results of the paper. Figure 6 shows more visualization results. We observe that the detection accuracy of the target using $Z_{com}$ is higher, and

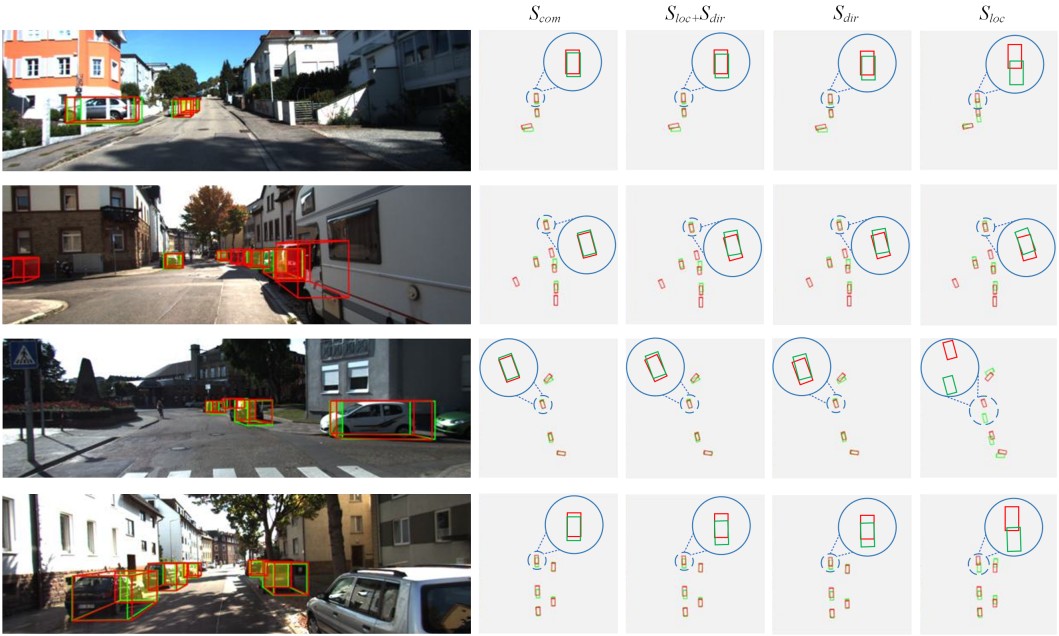

Figure 6: More visual results of 3D bounding boxes and BEV.

it has a satisfactory detection effect on small targets. This is because the operations of reconstructing depth and fusing depth through reliability scores improve the generalization ability of the estimated depth in different environments, making it more accurate when locating objects.

