# OpenReview forum: "VP-MonoMF: Visual Prompt Guided Monocular 3D Object Detection with Multiscale Fusion"
_ICLR.cc/2026/Conference — ICLR 2026 Conference Withdrawn Submission_

### Official Review · Reviewer_TRTi · 2025-10-29

**Soundness:** 2
**Presentation:** 1
**Contribution:** 2
**Rating:** 2
**Confidence:** 4

**Summary:**

This paper proposes a monocular 3D object detection method, VP-MonoMF, which mitigate the mutual interference between 2D and 3D detection branches by training them in separate stages. In addition, the method incorporates a depth reconstruction module to correct camera pose. To further enhance the detection of small objects, it employs a visual prompting mechanism that adaptively adjusts feature weights based on object size. The proposed approach achieves competitive performance on the KITTI 3D detection benchmark.

**Strengths:**

- The method demonstrates solid performance on the KITTI dataset.
- It introduces a visual prompting mechanism that adjusts feature weighting based on object size, improving detection accuracy for small objects.

**Weaknesses:**

1. **Questionable Motivation**
   - The claim that joint training of 2D and 3D detection branches leads to negative interference is not well justified.
   - Prior studies have shown that joint optimization can facilitate mutual benefits between 2D and 3D tasks.
   - The rationale for completely separating the two branches therefore appears unconvincing.

2. **Unclear Methodology and Contradictory Logic**
   - Several parts of the paper are vague and difficult to follow.
   - **Figure 1** lacks clarity: if the camera pose changes, both the depth map and the 3D bounding boxes should vary accordingly. As a teaser figure, it fails to highlight the core problem or main innovation.
   - The term **“3D depth information”** (Lines 71 and 73) is ambiguous, it is unclear whether it refers to estimated depth, ground-truth depth, or a learned feature representation.
   - The workflow is logically inconsistent: although the authors claim that 2D and 3D branches interfere with each other, the second stage trains the 2D detector using enhanced features derived from 3D training.
   - The decision to train **3D first, then 2D** is not adequately justified. The alternative order (2D → 3D) is not explored.

3. **Limited Novelty**
   - The paper reads more like a system description than a research paper. The overall design appears complicated rather than conceptually elegant. The proposed method primarily integrates several complex components (e.g., DLA, MDF, VPF) rather than introducing a fundamentally new concept.

4. **Poor Readability**
   - Numerous abbreviations (MDF, 3DR, VPF, etc.) are overused throughout the paper, reducing readability.  Subsection titles are vague and do not clearly convey their content.

5. **Overclaimed Results**
   - The abstract asserts state-of-the-art performance, but Table 1 shows that some results are not superior to existing approaches.

6. **Insufficient Evaluation**
   - Experiments are conducted solely on the KITTI dataset.  Evaluation on additional benchmarks such as nuScenes would be needed to validate the generalization capability.

**Questions:**

See weakness part.

---

### Official Review · Reviewer_2uNn · 2025-10-31

**Soundness:** 1
**Presentation:** 2
**Contribution:** 1
**Rating:** 0
**Confidence:** 5

**Summary:**

This paper introduces VP-MonoMF, a novel two-stage framework designed to improve accuracy and robustness for monocular 3D object detection. The proposed method consists of three main modules: 3D Depth Reconstruction (3DR), Multi-Depth Fusion (MDF), and Visual Prompt Fusion (VPF). The first stage trains the 3D branch including MDF, which estimates global depth map and local 3D depth information per object. The subsequent second stage trains the 2D detector using VPF module, which enhances the small object features by adaptively adjusting attention weights based on GT object size.

**Strengths:**

Unfortunately, none.

**Weaknesses:**

1. [Novelty] 3D object detection is a very mature task and field. There are many existing models and methods that perform much better than the proposed method, but they are neither cited nor referred.
2. [Benchmark] KITTI benchmark is a small and outdated dataset and no longer used in the field of 3D object detection. nuScenes is the minimum standard benchmark for 3D object detection.
3. [Performance Comparisons] The study does not compare with SOTA models.
3. [Method] Uses a combination of outdated methods and models. A complex method with limited performance.

**Questions:**

1. The introduced multi-task losses (equation 11, 15) appear to be a simple sum of their components, implying a uniform weight of 1.0 for each. Is this correct, or were different balancing weights (hyperparameters) used? If so, have you experimented with different weights?

---

### Official Review · Reviewer_BGc9 · 2025-10-31

**Soundness:** 3
**Presentation:** 4
**Contribution:** 3
**Rating:** 6
**Confidence:** 4

**Summary:**

1 The paper proposes VP-MonoMF, a two-stage monocular 3D object detection method that integrates Multi-Depth Fusion (MDF), 3D Depth Reconstruction (3DR) for camera pose correction, and Visual Prompt Fusion (VPF) to enhance small-object features using GT-guided adaptive attention.

2 It evaluates performance on the KITTI benchmark, reporting AP3D and APBEV at IoU=0.7 for Car under Easy/Mod/Hard settings, achieving SOTA results especially on Mod/Hard splits, and includes additional results on nuScenes.

3 Ablation studies validate each component (MDF, 3DR, VPF), depth fusion strategies, and the two-stage training scheme, demonstrating consistent improvements; experiments appear comprehensive and well-structured.v

**Strengths:**

The paper presents a well-motivated and technically sound approach to monocular 3D object detection, with clear strengths in originality—particularly through the novel integration of visual prompting guided by ground-truth object size, a two-stage training strategy to decouple 2D and 3D learning, and explicit handling of camera pose variation via 3D depth reconstruction. The method is thoroughly evaluated on KITTI and nuScenes, supported by comprehensive ablation studies that validate each component. The writing is clear, the problem formulation is realistic and addresses practical limitations of prior work, and the reported SOTA results—especially on small and hard objects—highlight its significance for real-world autonomous perception.

**Weaknesses:**

1 The paper’s reliance on ground-truth object size to generate the visual prompt limits its practical applicability, as GT is unavailable during inference.

2 The description of the Pose Detector and 3DR module lacks sufficient architectural and implementation details (e.g., how vanishing point estimation is trained or validated), making it hard to assess robustness or reproduce the camera pose correction.

3 While KITTI results are strong, experiments on nuScenes are only in supplementary material and lack per-category or small-object-specific metrics, leaving uncertainty about whether gains truly generalize beyond KITTI’s Car class and moderate/hard splits.

**Questions:**

Similar to Weakness.

---

### Note · Authors · 2025-12-17

**Comment:**

After discussion among the authors, it has been decided that this paper requires further refinement and will therefore be withdrawn.

**Withdrawal Confirmation:**

I have read and agree with the venue's withdrawal policy on behalf of myself and my co-authors.